# Temporal and Spatial Evolution of the African Swine Fever Epidemic in Vietnam

**DOI:** 10.3390/ijerph19138001

**Published:** 2022-06-29

**Authors:** Qihui Shao, Rendong Li, Yifei Han, Dongfeng Han, Juan Qiu

**Affiliations:** 1Key Laboratory of Monitoring and Estimate for Environment and Disaster of Hubei Province, Innovation Academy for Precision Measurement Science and Technology, Chinese Academy of Sciences, Wuhan 430077, China; shaoqihui18@mails.ucas.ac.cn (Q.S.); lrd@apm.ac.cn (R.L.); hanyifei21@mails.ucas.ac.cn (Y.H.); handongfeng@apm.ac.cn (D.H.); 2University of Chinese Academy of Sciences, Beijing 100049, China

**Keywords:** African swine fever, spatial analysis, spatial autocorrelation, spatiotemporal scan, Vietnam

## Abstract

African swine fever (ASF) is a severe infectious disease affecting domestic and wild suids. Spatiotemporal dynamics analysis of the ASF is crucial to understanding its transmission. The ASF broke out in Vietnam in February 2019. The research on the spatiotemporal evolution characteristics of ASF in Vietnam is lacking. Spatiotemporal statistical methods, including direction analysis, spatial autocorrelation analysis, and spatiotemporal scan statistics were used to reveal the dynamics of the spatial diffusion direction and spatiotemporal aggregation characteristics of ASF in Vietnam. According to the cessation of the epidemic, it was divided into three phases: February to August 2019 (phase 1), April to December 2020 (phase 2), and January 2021 to March 2022 (phase 3). The ASF showed a significant spread trend from north to south in phase 1. The occurrence rate of the ASF aggregated spatially in phase 1 and became random in phases 2 and 3. The high−high ASF clusters (the province was a high cluster and both it and its neighbors had a high ASF occurrence rate) were concentrated in the north in phases 1 and 2. Four spatiotemporal high-risk ASF clusters were identified with a mean radius of 121.88 km. In general, there were significant concentrated outbreak areas and directional spread in the early stage and small-scale, high-frequency, and randomly scattered outbreaks in the later stage. The findings could contribute to a deeper understanding of the spatiotemporal spread of the ASF in Vietnam.

## 1. Introduction

African swine fever (ASF) is a severe infectious disease caused by the African swine fever virus (ASFV) and it dominantly affects domestic and wild suids [1,2]. The ASF has high infectivity and mortality (close to 100%) [3,4]. It was listed by the World Organization for Animal Health (WOAH, founded as the OIE) as an animal disease that must be reported [5]. The ASFV has high resistance in the environment and long survival in various pork productions (ham, sausages, bacon, etc.) and on many materials (clothes, boots, wheels, etc.) [6,7,8]. The main routes of transmission of the ASF are direct contact with infected animals or their fluids or excretions and indirect contact through contaminated feed, pork productions, people, vehicles, or fomites [6,9,10]. Airborne transmission has been demonstrated over short distances [11]. Studies proved that the ticks (*Ornithodoros moubata*) and the fly (*Stomoxys calcitrans*) can transmit the ASFV [12,13]. The control of the ASF relied on culling infected pigs and restricting the transport of susceptible pigs before the commercial vaccine was available [14]. The ASF showed serious impacts on the production and breeding of domestic pigs, the survival of wild boars, as well as on the livelihoods of the farmers, national food security, and international trade [8,15,16].

The ASF originally emerged in East Africa in the early 1900s and has traditionally been present in the African continent [8,16,17]; the first introductions of the ASF to Europe were to Portugal in 1957 and again in 1960 [16]; in 2007, the disease was confirmed in the Caucasus region of Georgia. From there, the ASF virus gradually spread to neighboring countries (i.e., Armenia, Azerbaijan, Russia, and Belarus) [8]; in East Asia, the first outbreak of the ASF was reported in China in August 2018 [8,18], where the overall diffusion trend was from the northeast to the southwest [19]; since then, the ASF spread to 18 countries in Asia as of March 2022 [8]. Since more than 50% of the world’s pig population is in Asia, ASF outbreaks significantly affect the Asian and global pig industry [20]. 

Vietnam is the first country in Southeast Asia to report the ASF. In February 2019, the first ASF outbreak was officially reported in the Hung Yen province in Northern Vietnam and then spread to the entire country within 5 months [21]. There were no new ASF reports between September 2019 and March 2020. The ASF epidemic in Vietnam reappeared in April 2020 and continues to spread in 2021 and 2022. About six million pigs were culled from 2019 to 2021 [21,22]. Pig production contributes 60% of the total livestock output in Vietnam; the ASF outbreak caused severe economic losses among farmers and posed adverse impacts on the pig industry, leading to a decrease in the gross domestic product (GDP) and a substantial loss of jobs in Vietnam [21].

Gene sequencing can provide clues for tracing the origin of the epidemic; for example, studies showed that the ASFV isolates of the first outbreak in Vietnam belonged to the ASFV strains sequenced in Georgia and China [17,23]. However, on a small temporal and spatial scale, as the strains in Vietnam were of the same type within a short period, the method of gene sequencing will not be applicable. The spatiotemporal analysis method based on the disease maps of the epidemic is helpful for understanding its spread patterns, supporting the cause analysis of its spatiotemporal heterogeneity and control efforts [24,25,26]. For example, center feature analysis can visually reflect changes in the centers of disease distribution [27] and spatial autocorrelation analysis can identify whether the overall distribution of diseases is clustered or dispersed and detect the location of clusters and outliers [28,29]. Spatiotemporal scanning statistical analysis can detect high-risk clusters of disease in continuous time and space [30,31,32]. Lee et al. detected the high-risk clusters in the Lao Cai province, Vietnam [33]. To our knowledge, the spatiotemporal distribution characteristics of the ASF in the whole country of Vietnam since the outbreak remain unclear and it is worth further exploring. 

This study aimed to increase knowledge of the temporal and spatial evolution of the ASF in Vietnam, including its spreading direction and spatiotemporal clustering characteristics. 

## 2. Materials and Methods

### 2.1. Data and Sources

The ASF epidemic data in Vietnam from 1 February 2019 to 31 March 2022 were collected from the EMPRES-i Global Animal Disease Information System of the Food and Agriculture Organization of the United Nations (FAO) [34]. The information recorded included administrative division (commune-level), the observation date, the report date, and the location (latitude and longitude) of the outbreaks. An outbreak means the occurrence of one or more cases in an epidemiological unit (a group of animals with the same likelihood of exposure to a pathogenic agent) [5]. The Dictionary of Epidemiology defines an outbreak as an epidemic limited to a localized increase in the incidence of disease, e.g., village, town, or closed institution [35]. The epidemiological unit in this dataset is the commune (town district or ward). The observation date was the outbreak time of the ASF outbreaks. For the outbreaks with unknown observation dates (305/665), the report date minus the average days between the observation date and report date was used as the outbreak time.

The average ASF occurrence rate in each province (or municipality) per month was calculated using the following formula:(1)$$R_{i}=\frac{A_{i}}{C_{i}t}$$
where *R_i_* refers to the proportion of communes (town districts or wards) with ASF in the *ith* province (or municipality) per month in a given period, *A_i_* refers to the number of reported ASF outbreaks in the *ith* province in a period, *C_i_* refers to the number of communes of the *ith* province, and *t* is the number of months in the period. The numbers of communes in each province in 2019 and 2020 were obtained from the *Statistical Yearbook of Viet Nam* [36,37].

### 2.2. Spatial Statistical Methods

Based on the perspective and spatiotemporal scale of the spatiotemporal characteristics of the ASF epidemic in Vietnam to be revealed, we selected three spatial analysis methods (Table 1). In particular, the direction analysis was used to test the spatial transmission direction according to the monthly outbreak center; spatial autocorrelation analysis, including Global Moran’s *I* and Anselin Local Moran’s *I*, were used to reveal the clustering characteristics of different epidemic stages at the provincial level respectively; the spatiotemporal scanning statistics aimed to reveal the spatiotemporal high clusters at the daily outbreaks scale.

#### 2.2.1. Directional Analysis

Inaccurate outbreak timing may affect the accuracy of the direction analysis, for which it was decided to reveal possible directions on a monthly scale. Firstly, the monthly overall outbreak centers of the ASF were detected using the central feature analysis performed in the ArcGIS software (ESRI Inc., Redlands, CA, USA) [38]. Then the direction test was performed using ClusterSeer software (Biomedware Inc., Ann Arbor, MI, USA) [39] to determine whether the monthly outbreak centers of the ASF tended to be in a systematic, directional spread. 

The outbreak center was identified with the smallest distance accumulation from all other ASF outbreaks in a month. The distance accumulation was calculated by the Euclidean method and the formula was as follows:(2)$$D_{i}=\overset{n}{\underset{j=1,j\neq i}{\sum }}d_{ij}$$
where *D_i_* refers to the distance accumulation of the *ith* ASF outbreak and *d_ij_* refers to the distance between the *i*th outbreak and the *jth* outbreak. The central feature analysis requires three or more outbreaks. When there were less than three ASF outbreaks in a month, the first outbreak was taken as the outbreak center.

In the direction test, an infection chain was constructed based on the chronological order of the monthly ASF outbreak centers. Directed line segments were used to connect the outbreaks. The average angle and the angular concentration were calculated during the direction test. Angles are taken as rotated counterclockwise degrees from the horizontal, with East corresponding to 0 and North to 90. Concentration is in the range of 0 to 1, with 1 indicating an angular variance of 0. A consistent direction of the spread of cases will result in an angular concentration near 1. A random spread of cases will result in an angular concentration near 0. The time connection matrix was set as relative (each outbreak connected to all of the outbreaks that followed it). The significance of the test statistic was estimated by the Monte Carlo simulation (*p* < 0.05).

#### 2.2.2. Spatial Autocorrelation Analysis

Spatial autocorrelation analysis performed in the ArcGIS software was used to detect the presence of spatial aggregation in the ASF epidemic [28,29,40]. The global Moran’s *I* is calculated to analyze whether the ASF outbreaks demonstrate aggregation in the whole study area. The Anselin Local Moran’s *I* was used to find out the specific location of clusters or outliers [41,42]. The formulas of the global Moran’s *I* and the Local Moran’s *I* are:(3)$$I=\frac{n\sum_{i=1}^{n}\sum_{j=1}^{n}w_{ij}\left(x_{i}-\bar{x}\right)\left(x_{j}-\bar{x}\right)}{\left(\sum_{i=1}^{n}\sum_{j=1}^{n}w_{ij}\right)\sum_{i=1}^{n}{\left(x_{i}-\bar{x}\right)}^{2}}\;,\;i\neq j$$
(4)$$\;\;I_{i}=\frac{x_{i}-\bar{x}}{S_{i}^{2}}\overset{n}{\underset{j=1,j\neq i}{\sum }}w_{ij}\left(x_{j}-\bar{x}\right)$$
(5)$$S_{i}^{2}=\frac{\sum_{j=1,j\neq i}^{n}{\left(x_{j}-\bar{x}\right)}^{2}}{n-1}$$
where *I* refers to the global Moran’s *I*, *I_i_* refers to the local Moran’s *I* of the *ith* province, and $S_{i}^{2}$ refers to the sample variance of the ASF occurrence rate in provinces other than the *ith* province. *n* refers to the total number of the provincial administrative region, *x_i_* refers to the ASF occurrence rate of the *ith* province, *x_j_* refers to the occurrence rate of the *jth* province, and $\bar{x}$ refers to the average occurrence rate of the ASF in all provinces. *w_ij_* is the spatial weight of *x_i_* and *x_j_*, reflecting the spatial relationship between *x_i_* and *x_j_*. The inverse distance method was applied to define the spatial relationship between features such that nearby neighboring features have a larger influence on the target feature than features that are far away.

Global Moran’s *I* ranges from −1 to 1. The z-score and *p*-value are used to indicate the statistical significance of the calculated Moran’s *I* index values. The z-score represents multiples of the standard deviation and the *p*-value represents the probability of the observed spatial pattern being created by a random process. For example, when the z-score is greater than 2.58 or less than −2.58 and the *p*-value is less than 0.01, it indicates that the probability of random occurrence of this clustering pattern is less than 1% (99% confidence level). When the z-score or *p*-value indicates statistical significance, a positive Moran’s *I* index value indicates a tendency toward clustering while a negative Moran’s *I* index value indicates a tendency toward dispersion.

Local Moran’s *I_i_* > 0 indicates that the *ith* province is part of a cluster, surrounded by provinces with similarly high or low values. Local Moran’s *I_i_* < 0 indicates that the *i*th province is an outlier surrounded by provinces with dissimilar values. Based on *I_i_*, z-score and *p*-value, the provinces can be classified into five types: (1) High-high (HH) indicates a statistically significant cluster of high values; (2) Low-low (LL) indicates a statistically significant cluster of low values; (3) High-low (HL) indicates an outlier in which a high value is surrounded primarily by low values; (4) Low-high (LH) indicates outlier in which a low value is surrounded primarily by high values; (5) ‘not significant’ indicates no spatial autocorrelation.

#### 2.2.3. Retrospective Spatiotemporal Scan Analysis

Compared with the Local Moran’s *I* analysis revealing the spatial clusters or outliers of the epidemic within three phases, the retrospective spatiotemporal scan analysis detected ASF high-risk clusters in a continuous time range throughout the study period on a daily scale. The retrospective spatiotemporal scan analysis was performed using SaTScan software (Kulldorff, Cambridge, MA, USA) [43]. Based on the reported ASF outbreaks from 2019 to 2022, the space−time permutation model was applied to detect the spatiotemporal high-risk ASF clusters [44,45]. It is a probability model that is suitable for the detection of disease outbreaks when only the number of outbreaks is available and the expected number is calculated using only the outbreaks in the absence of excellent population-at-risk data [30].

The parameters necessary to run the model include the unique code, geographic location (longitude and latitude), and date of the ASF outbreaks. The analysis used a circular window to search the high-risk clusters. The scanning window moved in time and space to cover every possible geographic location and time interval. To ensure accuracy, days were used as the time unit to reflect the spread of the ASF. The maximum spatial cluster size was defined as 50% of the total number of ASF outbreaks and the maximum temporal cluster size was defined as 90 days [44]. The high-rate clusters were restricted to having at least 20 ASF outbreaks. The log-likelihood ratio statistic (LLR) was used to evaluate whether the scan window contains a clustered area and Monte Carlo simulations (999 times) were used to evaluate the significance of the detected clusters at a 0.05 level (*p*-value). The relative risk (RR) was the ratio of the number of observed ASF outbreaks in a cluster to the expected one, indicating the infection risk in a cluster compared to that in other areas [27,32].

## 3. Results

### 3.1. The Situation of the ASF in Vietnam

A total of 665 ASF outbreaks were reported in Vietnam from 1 February 2019 to 31 March 2022 (Figure 1). It can be divided into three phases according to the cessation of the epidemic, starting with (1) the first epidemic phase (from 1 February to 18 August 2019). There was a total of 198 ASF outbreaks in all 63 provinces (or municipalities) in this phase. The first ASF outbreak was observed in the Hung Yen province (Figure 2). The number of ASF outbreaks increased rapidly in a month. The first peak occurred in March with 116 ASF outbreaks and the second peak occurred in May with 41 ASF outbreaks. The Thai Binh province had the highest average occurrence rate at 1.97% and the national average rate was 0.22% in this phase (Figure 2). (2) The second epidemic phase was from 15 April to 21 December 2020. The ASF was first observed in the Bac Kan province, eight months after the end of the first wave. There was a total of 170 ASF outbreaks in 29 provinces (or municipalities). The first peak was in June with 29 outbreaks and the second was in October with 50 outbreaks. The Tuyen Quang province had the highest average occurrence rate at 2.25% and the national average rate was 0.19% in this phase. (3) The third epidemic phase was from 1 January 2021 to 31 March 2022. There was a total of 297 ASF outbreaks in 41 provinces (or municipalities). The ASF epidemic was characterized by long duration and repetition in this phase. The first ASF outbreak was in the Quang Nam province. The peak occurred in February 2022, with 70 ASF outbreaks. The Kon Tum province had the highest average occurrence rate at 1.11% and the national average rate was 0.20% in this phase.

### 3.2. The Directional Spreading of the ASF 

The direction test was statistically significant (*p* < 0.01) (Table 2). The average angle in phases 1, 2, and 3 was 278.5°, 264.85°, and 288.56°, respectively. In phase 1 (Figure 3a), the spreading center started in the Thai Binh province and ended in the Ninh Thuan province with an angular concentration of 0.74, indicating a high consistent southwardly direction of spread. In phases 2 and 3 (Figure 3b,c), the angular concentration was 0.35 and 0.18, respectively, indicating high angular variances (0.65 and 0.82, respectively), namely the random spread of monthly outbreaks.

### 3.3. Spatial Aggregation Characteristics of the ASF

#### 3.3.1. Spatial Autocorrelation Results of the ASF

The ASF occurrence rate in phase 1 had a significantly positive spatial correlation, indicating that the spatial distribution pattern was aggregated at the provincial level (*I* = 0.47, z-score = 8.14, *p* < 0.01) (Table 3). The spatial distribution pattern of the ASF epidemics showed randomness in phases 2 and 3 (*p* > 0.05).

Six provinces or municipalities were identified as HH clusters in phase 1: Bac Ninh, Ha Nam, Ha Noi, Hai Phong, Hung Yen, and Thai Binh (Figure 4a). Three provinces were identified as HH clusters in phase 2: Bac Kan, Ha Giang and Yen Bai (Figure 4b). There were no HH clusters in phase 3 (Figure 4c). All the HH clusters in phases 1 and 2 were located in Northern Vietnam (Figure 4a,b). Most of the LL clusters were concentrated in the south in the three phases. The Quang Tri province was an HL outlier in both phases 2 and 3 (Figure 4b,c).

#### 3.3.2. Spatiotemporal High-Risk Clusters

Space−time scan analysis identified four spatiotemporal high-risk clusters, with a total of 165 outbreaks, a mean radius of 121.88 km, and a mean duration of 45.5 days (Figure 1 and Figure 5, Table 4). All clusters were statistically significant (*p* < 0.01) and were sorted by LLR from high to low. With a radius of 46.76 km and a risk of outbreak 5.47 times higher than in other areas (*p* < 0.01), cluster 1 covered 79 outbreaks in five provinces (Thai Binh, Hung Yen, Ha Nam, Hai Duong, and Hai Phong) from 1 February to 8 March 2019 in phase 1 and concentrated in the Thai Binh province with 45 outbreaks. Cluster 2 covered 31 outbreaks in the Tuyen Quang, Yen Bai, and Bac Kan provinces from 16 September to 11 November 2019 in phase 2. Cluster 3 had the highest RR of 6.65 and the shortest duration of 12 days from 12–23 February 2022 in phase 3 and covered 26 outbreaks in five provinces (Dong Thap, Kien Giang, Soc Trang, Tien Giang, and Vinh Long). Cluster 4 had the largest radius of 254.01 km and the longest duration of 76 days from 25 January to 12 April 2021 in phase 3, covering 26 outbreaks in four provinces (Ha Tinh, Nghe An, Quang Nam, and Quang Tri). 

## 4. Discussion

### 4.1. Analysis of Temporal Characteristics of the ASF in Vietnam

The number of ASF outbreaks in March 2019 was much higher than in other months of 2020–2022. All outbreaks (100%) occurred at backyard pig farms with no or low biosecurity at the beginning [46]. Since the ASF had never appeared in Vietnam before 2019, the public lacked knowledge of the ASF and could not carry out effective prevention in the early stages of the epidemic [33,47].

The ASF was stopped seven months after the first outbreak. Necessary measures were implemented to control and stop the spreading of the ASF in Vietnam after the first detection: early detection, culling, disinfection, and compensation; strictly movement control of pig and pig products; biosecurity application in big farms; risk communication and public awareness; and information sharing and international collaboration [21,46].

The ASF reappeared in April 2020. The Vietnamese government endorsed the “National Plan for the Prevention and Control of African Swine Fever for the period of 2020–2025” on 7 July 2020, which set goals for ASF control, pig farm biosecurity application, and laboratory capacity development to be achieved; defined restocking conditions, sampling requirements, surveillance, conditions for culling, and moving-to-slaughter [48].

The ASF outbreak has swept across 63 provinces across the country in phase 1. After a nationwide outbreak, the number of provinces that have reported the ASF in phases 2 and 3 was 29 and 41, respectively (Figure 2). The average occurrence rate in phases 2 and 3 (0.19% and 0.20%, respectively) was less than that in phase 1 (0.22%). From the perspective of the distribution range and occurrence rate, it seems that the epidemic in the country has become less serious in phases 2 and 3 than that it was in phase 1. The ASF vaccine was expected to be announced in the second quarter of 2022 and used on large-sized pig farms (more than 300 pigs) in Vietnam [49]. However, as more than 90% of outbreaks occurred in small and medium-sized farms with poor biosecurity practices, this was still a great challenge for the prevention and control of the ASF [21].

The new ASF outbreaks decreased significantly in March 2022 compared to February. This may be related to the implementation of the Month of Environmental Sanitation and Disinfection aimed at actively preventing disease outbreaks in domestic animals in high-risk areas that started on 15 March 2022 [48]. The prevention and control measures of the ASF were refined and improved. In addition to implementing traditional measures such as comprehensive sanitation and disinfection, timely detection, and reporting and eradication of epidemic diseases, there were new measures including centralizing breeding farms, slaughter facilities, etc., and strengthening the capacity of the veterinary system [50].

There were seasonal differences in outbreaks of the ASF epidemics in Vietnam. From February to March in 2019, 2021, and 2022, the number of ASF outbreaks increased significantly (Figure 1). The increased human movements and the circulation of pork products during the Vietnamese New Year (in late January or early February) may have played an important role in the spread of the ASF [33]. Some peaks of the outbreak occurred in May 2019, June 2020, and May 2021 (Figure 1). Similar situations have occurred in Sardinia (Italy) and Poland, the ASF outbreak peaks of domestic pigs were in May and early summer, while the ASF outbreak peaks of wild boar were from October to February [31,45]. The sunny and windy weather conditions in autumn, winter, and spring proved to be conducive to the spread of the ASF virus in the air [51].

### 4.2. Analysis of Spatiotemporal Characteristics of ASF in Vietnam

In both phases 1 and 2, the ASF broke out first in the north, and the HH clusters and the high-risk cluster areas were both located in the north. The virus most likely entered the country from abroad through the illegal movement of pigs and pork products or food with the ASFV brought by international travelers to Vietnam [33,46,52]. Therefore, attention should be paid to the source of the virus and its spread in the northern region.

The overall ASF outbreak center changed from north to south in phase 1 (Figure 3). This transmission direction in the early stage of the introduction of the virus into Vietnam may be related to the transport of live pigs or feeds for swine and the movement of vehicles and people [21,46]. The concentration of the ASF spread direction decreased (Table 2) and the location of the ASF outbreak center seemed to become more and more disorderly in phases 2 and 3 (Figure 3). Similarly, the spatial autocorrelation analysis results also indicated the random and sporadic characteristics of the subsequent epidemic under the influence of various and complex natural and social−economic factors.

In phase 1 (February 2019–August 2019), there was a high-risk cluster from 1 February to 8 March 2019 (Figure 1), which roughly coincided with the HH cluster detected by local spatial autocorrelation (Figure 4a and Figure 5a). It was the Hung Yen province and its surrounding area where the initial outbreak occurred in Vietnam, indicating that the initial outbreaks developed rapidly and severely. The first outbreak in phase 2 was reported in the Bac Kan province. The Bac Kan province and its surrounding areas were detected as HH clusters and high-risk clusters (Figure 4b and Figure 5b). Similarly, the Quang Nam province and its surrounding areas were the origins of the ASF in phase 3, detected as high-risk clusters (Figure 5c). Our study confirmed the explosive infection at the beginning of each phase (high-risk clusters, HH cluster), which is likely to be related to the long incubation period (4–19 days) and rapid spread of African swine fever [53]. In addition, the lack of timely testing for the ASF and prevention and control measures at the beginning of the epidemic allowed more time for the virus to spread through the movement of infected pigs, pig products, vehicles, and people [6,33]. Therefore, we also appeal to the ASF testing for every animal presenting with high fever and death in high-risk areas to allow for early detection and to avoid the spread of explosive infection [53].

### 4.3. The Limitation of this Study

The limitation of this study is the uncertainty of the data. Typical animal disease outbreak information is usually obtained from the datasets published by two international agencies: the OIE’s WAHIS (https://wahis-uat.oie.int, accessed on 17 May 2022) and the FAO’s EMPRES-i (https://empres-i.apps.fao.org/, accessed on 17 May 2022). The differences between the two datasets are mainly reflected in the sources of data collection (member countries, official and unofficial reports, etc.), reporting methods (immediate and follow-up notifications, semester and annual reports, country or regional project reports, field mission reports, etc.), reporting content (animal health situations, exceptional epidemiological events, outbreaks of zoonotic disease, the number of animal outbreaks, percentages, prevalence and incidence) [24,54,55]. Consequently, there are differences in their spatiotemporal data. In the case of the H5N1, it showed similar trends in the temporal and spatial distributions if they were considered separately, but showed the differences in the joint spatiotemporal distribution [56]. Comparing the ASF outbreak data from the OIE and FAO, we found that the FAO’s data in Vietnam have more detailed spatiotemporal information and so they were chosen for this study. Inaccuracy of the data may lead to errors in spatiotemporal analysis results, especially in micro-spatiotemporal scales. In the future, we will investigate how to comprehensively use multi-source data and improve the quality of data cleaning and refining. At the same time, we also recommend more accurate and standardized data acquisition and sharing. Moreover, this study focused on the outbreak points and the outbreak occurrence rate. Due to the high infectivity of the ASFV [16] and mass culled pigs [21,22], the outbreak occurrence rate (the number of ASF outbreaks/the number of communes), rather than the actual pig’s infection rate (the number of infection pigs/the total number of pigs), can better reflect the actual state of the epidemic at the provincial scale.

## 5. Conclusions

The integrated use of multiple spatiotemporal analysis methods has been proven to be able to comprehensively reveal the spatiotemporal distribution characteristics of the epidemic from multiple perspectives studied in Vietnam. In general, there were significant concentrated outbreak areas and directional spread in the early stage of the introduction of the ASF virus into Vietnam and small-scale, high-frequency, and randomly scattered outbreaks in the later stage; this could contribute to a deeper understanding of the spatiotemporal spread of the ASF in Vietnam.

## Figures and Tables

**Figure 1 ijerph-19-08001-f001:**
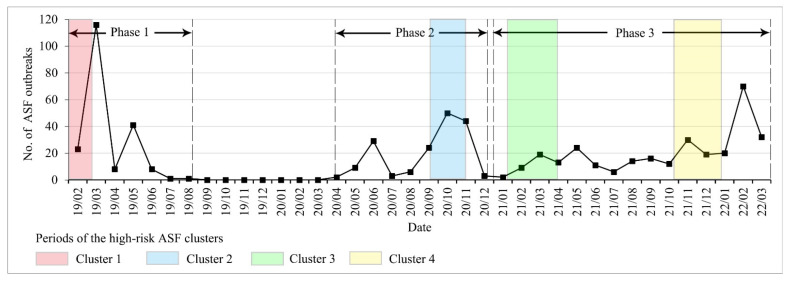
The number of new ASF outbreaks per month in Vietnam (February 2019–March 2022).

**Figure 2 ijerph-19-08001-f002:**
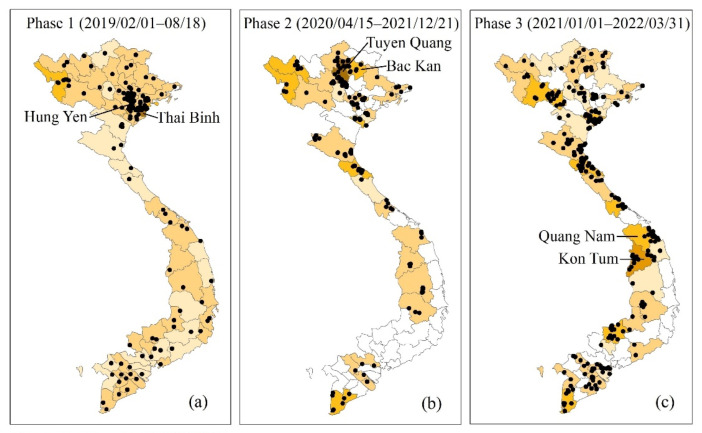
The ASF occurrence rate of each province in Vietnam (February 2019–March 2022). (**a**) Phase 1; (**b**) Phase 2; (**c**) Phase 3.

**Figure 3 ijerph-19-08001-f003:**
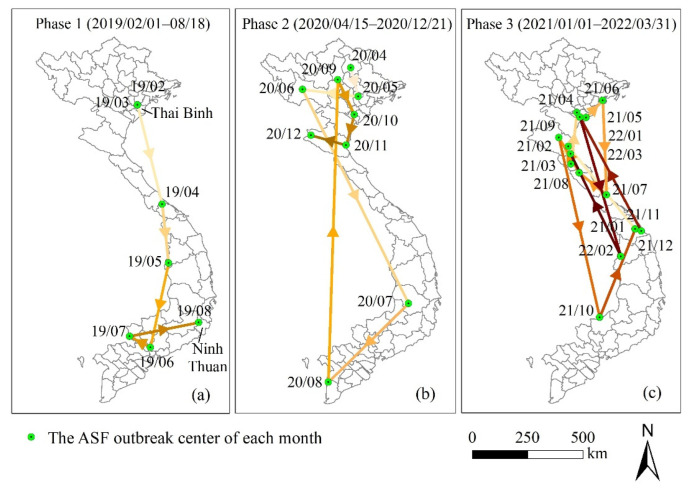
The monthly outbreak centers and average spread direction of the ASF (February 2019–March 2022).

**Figure 4 ijerph-19-08001-f004:**
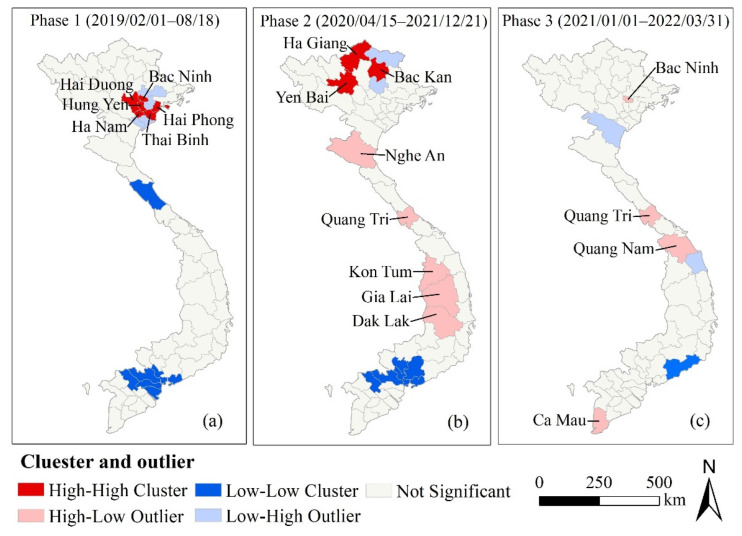
Maps of clusters and outliers of the ASF occurrence rate in Vietnam (February 2019–March 2022).

**Figure 5 ijerph-19-08001-f005:**
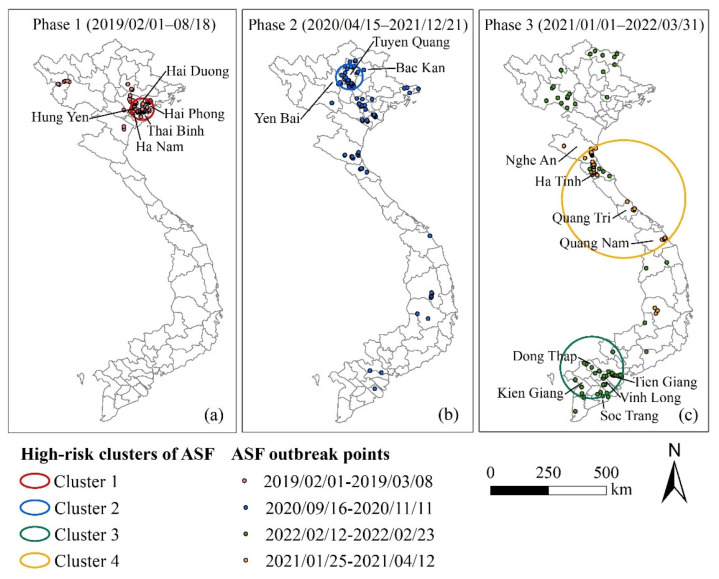
Map of spatiotemporal high-risk clusters of the ASF in Vietnam (February 2019–March 2022).

**Table 1 ijerph-19-08001-t001:** The comparison of spatial statistical methods used in this study.

Method		Data	Time Scale	Space Scale	The Revealed Spatiotemporal Characteristics
Direction analysis		Outbreak centers	Month	Outbreaks	Diffusion direction
Spatial autocorrelation analysis	Global Moran’s *I*	Occurrence rate	Phase	Province	Spatial distribution pattern (clustered, dispersed, or random)
	Anselin Local Moran’s *I*				Clusters or outliers
Retrospective spatiotemporal scan analysis		Outbreaks	Day	Outbreaks	Spatiotemporal high clusters

**Table 2 ijerph-19-08001-t002:** The directional test result of the ASF (February 2019–March 2022).

Phase	Date	No. of Central Points	Direction Test Result		
			Average Angle (°) *	Concentration	*p*-Value
1	February 2019–August 2019	7	278.51	0.74	0.001
2	April 2020–December 2020	9	264.85	0.35	0.001
3	January 2021–March 2022	15	288.56	0.18	0.001

* The average angle is rotated counterclockwise degrees from the horizontal, with East corresponding to 0 and North to 90.

**Table 3 ijerph-19-08001-t003:** Global Moran’s *I* of the ASF occurrence rate in Vietnam (February 2019–March 2022).

Phase	Date	Global Moran’s *I*			
		*I*	z-Score	*p*-Value	Distribution Pattern
1	February 2019–August 2019	0.47	8.14	<0.01	Aggregated
2	April 2020–December 2020	0.05	0.1.05	0.29	Random
3	January 2021–March 2022	−0.06	−0.58	0.56	Random

**Table 4 ijerph-19-08001-t004:** Spatiotemporal high-risk clusters of the ASF outbreaks in Vietnam (February 2019–March 2022).

Cluster	Date	Duration (Days)	Cluster Center Location	Radius (km)	Observed Points	Expected Points	RR ^1^	LLR ^2^	*p*-Value
1	1 January 2019–8 March 2019	36	20.676 N, 106.357 E	46.76	79	14.44	5.47	73.01	<0.01
2	16 September 2020–11 November 2020	58	21.972 N, 105.266 E	53.68	31	5.12	6.06	30.47	<0.01
3	12–23 February 2022	12	10.668 N, 105.561 E	133.07	26	3.91	6.65	27.54	<0.01
4	25 January 2021–12 April 2021	76	17.204 N, 106.798 E	254.01	29	5.41	5.36	25.53	<0.01

^1^ RR is the relative risk. ^2^ LLR is the log-likelihood ratio.

## Data Availability

The data of this work can be shared with the readers depending on the request.

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
