# Peer review of "Temporal and Spatial Evolution of the African Swine Fever Epidemic in Vietnam"

_ijerph, 2022, doi:10.3390/ijerph19138001_

Round 1
Reviewer 1 Report
check the attached file.

Author Response
Dear Reviewer:
Thank you for your approval of our manuscript entitled “Temporal and spatial evolution of African swine fever epidemic in Vietnam” (ID: ijerph-1754176). We tried our best to improve the manuscript and made some changes in the manuscript. These changes will not influence the framework of the paper and were marked in red in the revised manuscript. The whole article was carefully checked for spelling and grammar. The references are updated and checked that all references are relevant to the contents of the manuscript.
Yours Sincerely
Qihui Shao
Reviewer 2 Report
This study analyzed the epidemiology of ASF incidence in Vietnam using spatio-temporal epidemiological analysis. Although the study results may contribute to future ASF control in Vietnam, there are potential technical issues that need to be improved in this study. The English is easy to read and clearly conveys the authors' intentions.
Major comment
The introduction and discussion need to be reconsidered. With respect to the introduction, it should contain sufficient information necessary to understand the content of this study. Regarding the Discussion, the results obtained should be taken into consideration and fully discussed with comparisons to actual problems and other studies. Limitations and concerns in conducting this study should also be properly described.
Introduction
L36: Since the paper is about epidemiological analysis of ASF, it is necessary to explain the characteristics of the ASF virus and its mode of transmission. You will also need to explain the viability of the virus and its resistance to the environment. And you should also explain what is generally a problem with ASF in Asia, and here in particular with Vietnam. Doing so will enhance the significance of this study.
L39-40: I imagine that the spread of ASF in China has influenced the spread of ASF in Vietnam somewhat, and it would be good to include a brief description of how ASF spread throughout China after it was first identified.
L51-69: It is not necessary to list so many similar studies conducted in the past. Neither do we see any significance in including their details in the introduction. Rather, they should be cited as needed in the discussion. Instead, rather, please explain in more detail here the status and history of the ASF outbreak in Vietnam. You may also mention the overall ASF outbreak in Asia to date.
Materials and Methods
L83-92: Formula (1) involves many uncertainties. First, in OIE WAHIS, the number of reported outbreaks in Vietnam is more than 6000, which is significantly different from the FAO EMPRE-i results. Therefore, this discrepancy is likely to have a significant impact on the formula in (1).
Secondly, while most of the farms in Vietnam are backyard farms, there are also large corporate pig farms, thus the number of animals per farm varies. There might be some websites that mentions the percentage share of each farm size in Vietnam. I recommend authors to estimate the number of farms using a probability distribution.
These discrepancies and the importance of reliable information source need to be discussed in the Discussion.
L146-148: In my understanding, negative correlation means that it is spatially dispersed. Is it right?
L158-164: I guess “low cluster” means “observations are clustered, but not strongly autocorrelated”. If it so, it would be good to re-consider the description of each classification so as not to mislead.
L158-165: Outlier is a point which is significantly differs from other data groups. On the other hand, Moran’s I < 0 indicates observations are spatially more dispersed. Therefore, those two words contain different meanings. I recommend authors to change the words to avoid reader’s misunderstanding.
Results
Figure 3. This is a suggestion, but is it possible to change the color of the line and arrows over time from light to dark? It is difficult to see which direction the trend is moving when the same color is used.
Figure 5.
Other figures describe the results of the analysis for each phase. It would be easier to understand if the results of the spatio-temporal analysis were also depicted by dividing them into these three phases.
Discussion
The limitation in conducting this study has not been adequately explained. As mentioned earlier, the discrepancy in data at OIE and FAO should be a major obstacle to this study. Also, the number of farms is very limited in the method. Furthermore, despite the spatio-temporal epidemiological analysis, there is little description of the actual problems that may exist behind the results of each analysis (e.g., actual trade, illegal trade issues, backyard farmers, etc.). Spatio-temporal epidemiological analysis is more significant when considered in combination with the actual problems that are occurring. In Vietnam, several papers on epidemiological analysis, including spatio-temporal analysis, have already been published. A comparison of those results with the present results would enhance the significance of this paper.
L288-289: Sardinia is an island of Italy, therefore Sardinia and Italy are identical. Perhaps, authors wanted to indicate other country?
L298-300: I do not understand this sentence. Why have these actions by the Vietnamese government made it difficult to understand the actual infection rate of ASF?
Round 2
Reviewer 2 Report
Accept in present form
This manuscript is a resubmission of an earlier submission. The following is a list of the peer review reports and author responses from that submission.
Round 1
Reviewer 1 Report
This manuscript describes the spatiotemporal nature of ASF in Vietnam from 2019 to 2021. The manuscript is well written and organized and the scientific rationale and methods are sound. I do have a two suggestions that I believe will make the manuscript better; but this is a quality manuscript and is among the best I have reviewed.
- I think you need to include a limitations section. The methods you use while sound are quite simple compared to Bayesian space - time techniques. For example, I wonder when you aggregate your case counts by zone, which it appears you have to do for the local Anselin and space scan statistic if you are only measuring the number of swine in those zones. I realize that it would be next to impossible to collect that number to create a rate but how might this impact your findings or results? Maybe there are more cases in North Vietnam simply because there are more swine there.
- When discussing using ArcGIS in your methods I would not state you used "EUCLIDEAN_DISTANCE" like it is presented in the software. I think simply stating you used Euclidean as your distance specification is better.
Author Response
Dear Reviewers:
Thank you for your comments concerning our manuscript entitled “Temporal and spatial evolution of African swine fever epidemic in Vietnam” (ID: ijerph-1650401). Those comments are all valuable and very helpful for revising and improving our paper, as well as the important guiding significance to our research. We have studied the comments carefully and have made corrections which we hope to meet with approval. The main corrections in the paper and the response to the reviewer’s comments are as flowing:
Point 1: I think you need to include a limitations section. The methods you use while sound are quite simple compared to Bayesian space - time techniques. For example, I wonder when you aggregate your case counts by zone, which it appears you have to do for the local Anselin and space scan statistic if you are only measuring the number of swine in those zones. I realize that it would be next to impossible to collect that number to create a rate but how might this impact your findings or results? Maybe there are more cases in North Vietnam simply because there are more swine there.
Response 1: The research method we used sounds simple but can intuitively reflect the spatial variation characteristics of ASF, and has been applied in ASF spatial analysis in many countries and regions (lines 58-73). The statistical heads of pigs by province are added to estimate the number of farms in each province. The incidence rate of ASF was calculated by the number of ASF outbreak points and the farms when performing the spatial autocorrelation analysis (lines 95-1012). Bias caused by differences in the infected ASF cases and the scale of pig farms between regions may be ignored. The limitations section has been added in the discussion part (lines 369-377).
Point 2: When discussing using ArcGIS in your methods I would not state you used "EUCLIDEAN_DISTANCE" like it is presented in the software. I think simply stating you used Euclidean as your distance specification is better.
Response 2: Line 114, “the ‘EUCLIDEAN_DISTANCE’ method” was corrected as “the Euclidean method”.
We tried our best to improve the manuscript and made some changes in the manuscript. These changes will not influence the framework of the paper. And here we did not list the changes but marked them in red in the revised paper.
We appreciate for Editors/Reviewers’ warm work earnestly and hope that the correction will meet with approval.
Once again, thank you very much for your comments and suggestions.
Yours Sincerely
Qihui Shao

Reviewer 2 Report
Brief Summary
Authors used spatiotemporal statistical methods, including central feature analysis, spatial autocorrelation analysis, and spatiotemporal scan statistics to analyze the spatiotemporal evolution characteristics of African swine fever endemic in Vietnam from 2019 to 2021. Data were collected from the EMPRES-I Global Animal Disease Information System of the Food and Agriculture Organization of the United Nations (FAO). Using this data, that included specific time, latitude and longitude, and type of diseased animal from the outbreak points, they calculated the number of reported ASF outbreak points in each province.
Authors state that the analyzed data could assist with decision making support for the government to identify epidemic prevention and control areas and to scientifically improve epidemic prevention and control strategies. The issue however is that this is a retrospective analysis, therefore the authors to not articulate how this could assist with ongoing outbreaks if they data is only analyzed after the fact. Additionall, they do not provide how the data could be disseminated to assist with prevention and control strategies.
A total of 538 ASF outbreak points were reported in Vietnam over the study time period. Three points were wild boar, two in captivity, and one in the wild, all the others were domestic swine. Were the wild boar outbreaks responsible for the domestic swine infections? Were these the reservoir for the virus? Did it play any role? Authors appear to focus only on the statistics of the numbers and do not try to bring to the practical level.
Authors state location of ASF clusters changed every year indicating that these provinces have implemented effective ASF prevention and control measures – how do they come to that conclusion? What information was used? – statistics done does not tell one that.
Authors state that retrospective spatiotemporal scan analysis can dissect the actual spread rather than the overview of apparent continuous spread, but do not show how this “after the fact” information can help to predict the disease spread. Authors should expand on how they think the retrospective analysis can improve and lessen the impact of ASF.
Authors state there were seasonal differences and then state the analysis showed clusters could occur at any time. There appears somewhat contradictory and therefore would need explaining
Authors make general statements about meteorological factors, climate indicators; natural environment can all play a role but do not tie all together and make suggestions or provide support for how they can be used to assist with prevention and control
Authors state the study could be improved – but don’t provide how or what.
They state results would be more accurate if they had more data on specifics? How do they plan to obtain that data? Could they assist with the data collection? Could they make recommendations to the appropriate authorities, play an active role?
Specific comments
Introduction: high-high and low-low – recommend authors provide definition for those that only read the abstract and are not familiar with methodology
Line 134 – add in captivity
Author Response
Dear Reviewers:
Thank you for your comments concerning our manuscript entitled “Temporal and spatial evolution of African swine fever epidemic in Vietnam” (ID: ijerph-1650401). Those comments are all valuable and very helpful for revising and improving our paper, as well as the important guiding significance to our research. We have studied the comments carefully and have made corrections which we hope to meet with approval. The main corrections in the paper and the response to the reviewer’s comments are as flowing:
Point 1: The study is only based on case data.
For a thorough analysis, the Vietnamese ASF surveillance system must be described and the risk of missing ASF outbreaks (detection failure, under-reporting etc.) needs to be assessed. A case definition is missing. It remains unclear, how many pig holdings were checked for ASF and which proportion of them suffered an ASF outbreak (prevalence, incidence).
Response 1: The study focus on the spatiotemporal characteristics of the ASF outbreak points (lines 87-91). The statistical heads of pigs by province are added to calculate the incidence rate of ASF when performing the spatial autocorrelation analysis (lines 95-102). According to the OIE, the definition of an outbreak of the infectious disease was ‘the occurrence of one or more cases in an epidemiological unit’, where a case means ‘an individual animal infected by a pathogenic agent, with or without clinical signs’ [16]. About 6 million pigs were culled in Vietnam in the past three years[8, 38]. The exact number of pigs infected with the ASF virus was difficult to determine because some pigs might be culled before they could be tested. Therefore, we analyzed the reported ASF outbreak points data which was the most available and reliable. This limitation was explained in the Discussion section (lines369-377).
Point 2: The methods used to analyse the case data are not adequately described. It is therefore not possible to scrutinize the methods and the results. You claim, for example, that you used SaTScan (space-time permutation scan statistics). In my view, this may lead to bias, since the population has probably changed due to the influence of ASF, so that the outcome of the analysis may be heavily biased. By the way, SaTScan should be appropriately cited.
Response 2: The principle of space-time scan analysis is complex, so we briefly introduced how to use this method in the paper, and more details can be obtained from the reference [28, 29] (lines 166-185). Since a huge amount of pigs were culled since the ASF outbreak, accurate data on the population was unavailable. Therefore, we used space-time permutation scan statistic for the detection of ASF outbreaks that uses only outbreak point numbers, with no need for population-at-risk data. This model makes minimal assumptions about the time, geographical location, or size of the outbreak, and it adjusts for natural purely spatial and purely temporal variation [29]. The citation of the source of the SaTScan software is added on line 105.
Point 3: Also the "Central Feature analysis" is not sufficiently described. It remains open, how the data were analysed and if the results are scientifically sound.
Response 3: The description and the formula of distance accumulation of "Central Feature analysis" are added on lines 110-117. The ASF Center was previously calculated by years, and now it is calculated by month. This helps to clearly show the spread direction of ASF (lines 214-234).
Point 4: The same holds true for the "Spatial autocorrelation analysis". How was this done? Software? Algorithm? Settings?
Response 4: The spatial autocorrelation analysis was performed in ArcGIS software (lines 106-108). The description and the formula of Spatial autocorrelation analysis are added on lines 125-137.
We tried our best to improve the manuscript and made some changes in the manuscript. These changes will not influence the framework of the paper. And here we did not list the changes but marked them in red in the revised paper.
We appreciate for Editors/Reviewers’ warm work earnestly and hope that the correction will meet with approval.
Once again, thank you very much for your comments and suggestions.
Yours Sincerely
Qihui Shao

Reviewer 3 Report
The study is only based on case data.
For a thorough analysis, the Vietnamese ASF surveillance system must be described and the risk of missing ASF outbreaks (detection failure, under-reporting etc.) needs to be assessed. A case definition is missing. It remains unclear, how many pig holdings were checked for ASF and which proportion of them suffered an ASF outbreak (prevalence, incidence).
The methods used to analyse the case data are not adequately described. It is therefore not possible to scrutinize the methods and the results. You claim, for example, that you used SaTScan (space-time permutation scan statistics). In my view, this may lead to bias, since the population has probably changed due to the influence of ASF, so that the outcome of the analysis may be heavily biased. By the way, SaTScan should be appropriately cited.
Also the "Central Feature analysis" is not sufficiently described. It remains open, how the data were analysed and if the results are scientifically sound.
The same holds true for the "Spatial autocorrelation analysis". How was this done? Software? Algorithm? Settings?
Author Response
Dear Reviewers:
Thank you for your comments concerning our manuscript entitled “Temporal and spatial evolution of African swine fever epidemic in Vietnam” (ID: ijerph-1650401). Those comments are all valuable and very helpful for revising and improving our paper, as well as the important guiding significance to our research. We have studied the comments carefully and have made corrections which we hope to meet with approval. The main corrections in the paper and the response to the reviewer’s comments are as flowing:
Point 1: The study is only based on case data.
For a thorough analysis, the Vietnamese ASF surveillance system must be described and the risk of missing ASF outbreaks (detection failure, under-reporting etc.) needs to be assessed. A case definition is missing. It remains unclear, how many pig holdings were checked for ASF and which proportion of them suffered an ASF outbreak (prevalence, incidence).
Response 1: The study focus on the spatiotemporal characteristics of the ASF outbreak points (lines 87-91). The statistical heads of pigs by province are added to calculate the incidence rate of ASF when performing the spatial autocorrelation analysis (lines 95-102). According to the OIE, the definition of an outbreak of the infectious disease was ‘the occurrence of one or more cases in an epidemiological unit’, where a case means ‘an individual animal infected by a pathogenic agent, with or without clinical signs’ [16]. About 6 million pigs were culled in Vietnam in the past three years[8, 38]. The exact number of pigs infected with the ASF virus was difficult to determine because some pigs might be culled before they could be tested. Therefore, we analyzed the reported ASF outbreak points data which was the most available and reliable. This limitation was explained in the Discussion section (lines 369-377).
Point 2: The methods used to analyse the case data are not adequately described. It is therefore not possible to scrutinize the methods and the results. You claim, for example, that you used SaTScan (space-time permutation scan statistics). In my view, this may lead to bias, since the population has probably changed due to the influence of ASF, so that the outcome of the analysis may be heavily biased. By the way, SaTScan should be appropriately cited.
Response 2: The principle of space-time scan analysis is complex, so we briefly introduced how to use this method in the paper, and more details can be obtained from the reference [28, 29] (lines 166-185). Since a huge amount of pigs were culled since the ASF outbreak, accurate data on the population was unavailable. Therefore, we used space-time permutation scan statistic for the detection of ASF outbreaks that uses only outbreak point numbers, with no need for population-at-risk data. This model makes minimal assumptions about the time, geographical location, or size of the outbreak, and it adjusts for natural purely spatial and purely temporal variation [29]. The citation of the source of the SaTScan software is added on line 105.
Point 3: Also the "Central Feature analysis" is not sufficiently described. It remains open, how the data were analysed and if the results are scientifically sound.
Response 3: The description and the formula of distance accumulation of "Central Feature analysis" are added on lines 110-117. The ASF Center was previously calculated by years, and now it is calculated by month. This helps to clearly show the spread direction of ASF (lines 214-234).
Point 4: The same holds true for the "Spatial autocorrelation analysis". How was this done? Software? Algorithm? Settings?
Response 4: The spatial autocorrelation analysis was performed in ArcGIS software (lines 106-108). The description and the formula of Spatial autocorrelation analysis are added on lines 125-137.
We tried our best to improve the manuscript and made some changes in the manuscript. These changes will not influence the framework of the paper. And here we did not list the changes but marked them in red in the revised paper.
We appreciate for Editors/Reviewers’ warm work earnestly and hope that the correction will meet with approval.
Once again, thank you very much for your comments and suggestions.
Yours Sincerely
Qihui Shao

Round 2
Reviewer 3 Report
My main concern is still that only outbreak data were used. As long as this problem is not not resolved, your analysis is incomplete as appropriate denominator data are missing. It may even lead to false interpretations and invalid conclusions.